# Piloting the Schistosomiasis Practical and Precision Assessment approach in five health districts of the N'zérékoré region, Republic of Guinea

Balla Moussa Keita[ID][1][☺]*, Mamadou Diallo[1☺], Mariam Diarra[2☺], Souleymane Traore[1☺], Antoine Tamba Kamano[2☺], Dienabou Keita[1], Ablam Amento[1], Moussa Sylla[3], N'falaye Kante[4], Alber Dopavogui[4], Cheick Mouctar Sylla[4], Sekou Berete[1], Penelope Vounatsou[5], Stella Kepha[6], Katherine Gass[7☺], Rachel L. Pullan[8☺], Fiona M. Fleming[9☺], Nouhou Konkoure Diallo[2☺], Mandy Kader Konde[1☺]

1 Health and Sustainable Development Foundation (FOSAD) in French, Conakry, Guinea, 2 National Program for the Fight Against Neglected Tropical Diseases with Preventive Chemotherapy (PNL-MTN-CTP), Conakry, Guinea, 3 Mafrinyah Rural Health Training and Research Center, Mafrinyah, Guinea, 4 Polyclinique Communautaire Moderne (PCM), Dubreka, Guinea, 5 Swiss Tropical and Public Health Institute, Allschwil, Switzerland, 6 Kenya Medical Research Institute, Nairobi, Kenya, 7 Task Force for Global Health, Decatur, Georgia, United States of America, 8 London School of Hygiene and Tropical Medicine, London, United Kingdom, 9 Unlimit Health, London, United Kingdom

☺ These authors contributed equally to this work.
* keita.ballamoussa@gmail.com

## Abstract

### Background

In Guinea, N'Zérékoré region has historically been endemic for both *Schistosoma mansoni* and *S. haematobium*. Following eight years of mass treatment with praziquantel to treat schistosomiasis, as part of a multi-country project, the country was selected to pilot the Schistosomiasis Practical and Precision Assessment (SPPA) approach. The SPPA pilot was conducted in five health districts in the forest region. The main objectives were to determine the current infection status and treatment strategy for each health sub-district and to evaluate the feasibility of the SPPA approach.

### Methodology/Principal findings

A cross-sectional study among children aged 10–14 years of age was conducted. In each health district, a systematic sample of 15 schools were selected with 32 school children selected randomly from each. Stool and urine samples were collected from each child. Two Kato-Katz slides were examined for *S. mansoni* and soil transmitted helminthiasis (STH) and one urine filtration slide and one hemastix for *S. haematobium* infections and microhaematuria, respectively.

Of the 2400 children targeted for inclusion, 2325 provided samples (96.9%). The combined prevalence of *Schistosoma* species across the five health districts was

**Data availability statement:** All relevant data from this study has been uploaded to the publicly available COR-NTD Dataverse site and can be accessed directly at this link: https://doi.org/10.15139/S3/BY25HS Questions about the data availability can be sent to ntdsc@taskforce.org.

**Funding:** This work received financial support from the Bill & Melinda Gates Foundation and from the United States Agency for International Development (USAID) through its Neglected Tropical Diseases Program through their support of the Coalition for Operational Research on Neglected Tropical Diseases (COR-NTD) grant (Coordinator: BMK, Principal investigator: MKK,). COR-NTD is funded at The Task Force for Global Health primarily by the Bill & Melinda Gates Foundation and USAID. The funders had no role in the study design, data collection and analysis, decision to publish or preparation of the manuscript.

**Competing interests:** The authors have declared that no competing interests exist.

66.4%. *S. mansoni* had a high prevalence of 66.1% with four health districts above 50%. *S. haematobium* had a low prevalence of 4.3%. The overall prevalence of any combined STH (*Ascaris lumbricoides, Trichuris trichiura or hookworm*) was 11.7%. Sex, age and contact with a freshwater body during the last week before the survey, were not statistically significant in their association with schistosomiasis.

## Conclusion

The results of the SPPA indicate that schistosomiasis remains homogeneously high across all five health districts. Consequently, it is recommended to maintain annual treatment in each sub-health district, and to extend treatment to whole communities aged two years of age and over, while strengthening critical cross-sectoral interventions such as behaviour change and environmental management.

## Author summary

The manuscript presents a pilot study of the Schistosomiasis Practical and Precision Assessment (SPPA) approach in five health districts of the N'zérékoré region, Republic of Guinea. The SPPA is an evidence-based approach for the monitoring and evaluation of schistosomiasis programme progress through impact assessments. The study aimed to pilot and test the SPPA approach in terms of its feasibility when implemented by national programmes and its ability to support classification of sub-districts according to treatment needs. The results of this study will provide crucial information to improve schistosomiasis control strategies and optimize precision public health interventions in endemic regions.

## Introduction

Schistosomiasis, or bilharzia, is a parasitic disease caused by schistosome trematode infection [1]. The current strategy recommended by the World Health Organization (WHO) for most countries is to first control the morbidity associated with parasitic infections, before aiming to eliminate the disease as a public health problem. This is done through preventive chemotherapy (PC) with praziquantel (PZQ), targeting people aged two years and above [2,3]. The majority of schistosomiasis endemic countries in sub-Saharan Africa have successfully scaled up PC with PZQ, either in schools and/or communities in health districts where prevalence of infection exceeds 10%. As a result, the epidemiological profile of schistosomiasis is expected to have changed significantly, justifying the need to reassess the prevalence of schistosomiasis after several years of mass drug administration [2,3]. The WHO recommends that schistosomiasis control programmes that have conducted at least five rounds of PC with ≥75% coverage should conduct impact assessments to determine whether treatment frequency should be adjusted or maintained [2–4]. Although impact

assessments are necessary for neglected tropical disease (NTD) programmes to evaluate the impact of mass treatment with PC on diseases, such as schistosomiasis, WHO has not yet provided clear evidence-based guidance on how to efficiently conduct these assessments [5–8]. Another challenge arises from the fact that schistosomiasis programmes aspire to shift the implementation of PC from the health district to the sub-district level, with the aim of improving treatment targeting given the focal distribution and transmission of this infectious disease [5,6,8].

To address this programmatic gap, the "Schistosomiasis Oversampling Study" (SOS) was conducted between 2021 and 2023 to identify the optimal sampling method for impact assessments which would be feasible, cost-effective, and allow for accurate classification of sub-districts according to treatment needs [9,10]. In May 2023, SOS study teams, regional programme managers, international experts, non-government organisation (NGO) partners, donors, and WHO met in Nairobi, Kenya, to analyse the results. They reviewed different sampling strategies and agreed on a single approach called "Schistosomiasis Practical and Precision Assessments (SPPA)" [10]. This strategy was then piloted across selected countries to assess its feasibility and cost. Lessons learned from the pilots were intended to improve the SPPA protocol and training tools, for subsequent sharing and uptake by schistosomiasis programmes through the WHO's Expanded Special Project for Elimination of Neglected Tropical Diseases (ESPEN) [11].The Republic of Guinea was selected to pilot the SPPA approach in five health districts of the forest region due to the existence of: i) endemic health districts that had benefited from more than five effective rounds of PC and ii) good expertise in the country to conduct routine monitoring and evaluation surveys for schistosomiasis.

Guinea is endemic for both intestinal (*Schistosoma mansoni*) and urogenital schistosomiasis (*S. haematobium*) [12,13]. Early surveys conducted in 1995 as part of government-led school health and nutrition activities indicated prevalence ranged between 19.9% and 25.0% across the country [14]. Whilst more recently in 2010, surveys in Beyla and Macenta suggested prevalence of *S. mansoni* as high as 66.2%, *S. haematobium* 21%, *Ascaris lumbricoides* 8.1%, hookworm 51.2%, and *Trichuris trichiura* 2.4% [15]. More comprehensive mapping was conducted in 2014 to inform the roll out of the annual PC programme. This mapping used a purposive sampling method, covering 5 sites per health district. It identified a total of 31 health district endemic with schistosomiasis, including: 12 hyper-endemic districts (with a prevalence ≥50%), nine meso-endemic health district (prevalence ≥10% and <50%) and 10 hypo-endemic health district (prevalence ranging from 1 to 9%) [12,13].

The strategy to combat these diseases used the distribution of PC through mass drug administration (MDA), which was led by the National PC-NTD Control Programme under the Ministry of Health and Public Hygiene of Guinea, started in 2014 with Guéckédou health district and scaled up in 2016 to all endemic health districts. The PC was delivered according to the endemicity status of the health district for schistosomiasis and soil-transmitted helminths (STH) and based on the WHO guidance [4,16]. MDA campaigns with PZQ and albendazole or mebendazole, for schistosomiasis and the STH, respectively, were conducted in the community, targeting children aged 5–14 years. Across all endemic districts there had been five to eight rounds of MDA to the target age-group. The most recent MDA, prior to the SPPA pilot, was carried out in September 2023 and the endemic health districts obtained a reported programmatic coverage of between 75 and 100%. Hence, there was a need to reassess the epidemiological profile of schistosomiasis in these historically hyper-endemic health districts and assess the impact of the PC programme. The aim of the pilot study was to determine the current infection status and treatment strategy for each health sub-district and to assess the feasibility of this new SPPA approach to schistosomiasis impact assessments. The hypothesis going into this study is that the prevalence of schistosomiasis in school aged children will have decreased significantly following 5–8 rounds of preventive chemotherapy.

## Materials and methods

### Ethics statement

This study received administrative authorization from the Ministry of Health and Public Hygiene of Guinea and approval from the National Committee for Health Research Ethics (NCHRE 053/CNERS/24). The study was carried out by the Health and Sustainable Development Foundation (FOSAD) under the supervision of the National PC-NTD Control

Programme. On-site information and advocacy meetings were organised in collaboration with health authorities at all levels to obtain informed consent from village chiefs (verbal), parents or legal guardians of children (written), and school authorities (written), as well as assent from children (written). All positive children were treated in accordance with the guidelines of the National PC-NTD Control Programme.

## Study setting and population

The study was conducted across five health districts in the Forest Region of Guinea: Beyla (population: 326,082), Guéckédou (population: 290,611 and 14 sub-health districts), Lola (population: 171,561 and nine sub-health districts), N'Zérékoré (population: 396,949 and 15 sub-health districts) and Yomou (population: 114,371 and seven sub-health districts) [13]. These all had high prevalence of schistosomiasis during baseline mapping conducted in 2014 [10]. This region in the south of the country, bordered by Côte d'Ivoire, Liberia and Sierra Leone, is densely forested with many permanent water bodies (marshes, rivers and swampland) and a long rainy season. In recent years, there has been considerable deforestation for agriculture, driven in part by an influx of refugees fleeing conflicts in neighbouring countries. Primary occupations include rice cultivation, fishing and game hunting. Schistosomiasis transmission is perennial and thought to be linked to the many irrigation canals [14]. There has been an annual PC with PZQ delivered to school-age children and/or communities with high coverage. The most recent round, in 2023, was reported to have achieved coverage rates exceeding 96% in all study districts.

## Study design and sample size

Following the SPPA guidance, the National PC-NTD Control Programme reviewed the data and information for the area and determined that due to historically homogeneously high prevalence and other factors, a Practical Assessment would be the initial survey, with Precision Assessments being conducted later, if required (Fig 1) [10]. The Practical Assessment calls for a cross-sectional study of school-age children, aged 10–14 years, to determine the prevalence and intensity of schistosomiasis and STH. A two-stage cluster sampling process was carried out, with schools forming the primary sampling unit. The first stage used the Excel tool known as the "Practical Assessment Systematic Sampling Tool" for the systematic random selection of 15 schools per health district from an exhaustive list of schools in each health sub-district [11]. The second stage consisted of the random selection of 30 school-age children in each school. To account for children who chose not to provide a sample or were unable to do so, two additional children were invited to participate from each school. Therefore, in each school, 32 school-age children (16 girls and 16 boys) were randomly selected from grades three to six in the selected primary schools. Any child who was not in the target age group (10–14 years of age), was ill, did not give their assent, or whose parents did not consent were excluded from the study.

A total of 2400 school-age children were targeted for selection across the 75 schools in the five health districts concerned, thus in each health district, 480 school-age children from across 15 schools would be sampled. This sample size is based on the results of thousands of simulations run across transmission archetypes in six different countries and was found to maximize the number of times the sub-district was correctly classified in the SOS study [10].

## Collection, analysis and interpretation of data and samples

Prior to data collection, two field teams each consisting of two experienced laboratory technicians, a data enumerator and a supervisor were trained for three days including one day of piloting. Data were recorded electronically through tablets using the standardised Practical Assessment form provided by ESPEN and hosted on the ESPEN platform via the ESPEN Collect application [17].

Socio-demographic information of children (gender, age), risk of exposure, i.e., contacts with fresh water, as well as the GPS coordinates of each school, were collected. A stool and urine sample were collected from each child

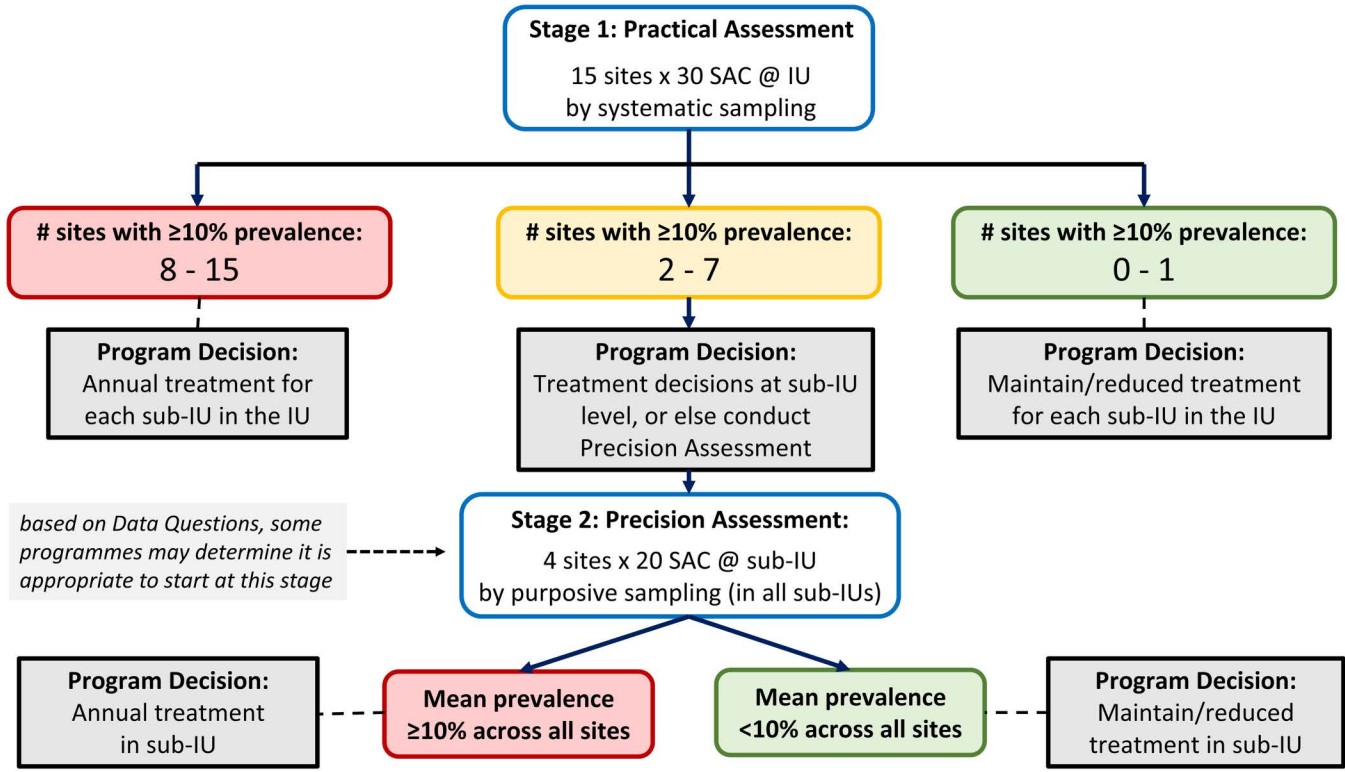

**Fig 1. SPPA decision tree.** implementation unit (IU), typically corresponds to a district, county, woreda, cercles, zone de santé, commune etc. In the context of Guinea IU is the health district; sub-IU, a smaller administrative area within an implementation unit, the new unit for schistosomiasis preventive chemotherapy. In Guinea, sub-IU is the health sub-district.

between 10:00 and 14:00. Two Kato-Katz slides were examined for each stool sample to detect *S. mansoni* and STH infection. Hemastix strips were used for detection of microhaematuria and urine filtration (one slide) was conducted for *S. haematobium* egg counting on all urine samples. The samples were prepared and analysed on site within the first hours after collection. When the analysis conditions at the school site were not favourable, samples were transported to the laboratory of the sub-district health centre, where they were prepared and analysed within four and five hours of collection.

Any school-age child with at least one egg in their sample (stool and/or urine) was considered positive for schistosomiasis and/or STH depending on the egg identified. For stool samples, infection intensity was calculated using the number of eggs per gram of stool according to the Kato-Katz method by multiplying the mean egg count of two slides by 24. Thus, the intensity of infection was classified as: high (≥400 epg), moderate (100–399 epg) or low (<100 epg) for *S. mansoni*; high (≥4,000 epg), moderate (2,000–3,999 epg) or low (<2,000 epg) for hookworm; high (≥50,000 epg), moderate (5,000–49,999 epg) or low (<5,000 epg) for *A. lumbricoides*; high (≥10,000 epg), moderate (1,000–9,999 epg) or low (1–9,999 epg) for *T. trichiura*. Infection intensity for *S. haematobium* was classified for each child as high (≥50 eggs/10 ml) or low (<50 eggs/10 ml) [16].

Data were cleaned and analysed using R software (v 4.4.0; R Core Team 2024). Maps were created using QGIS Desktop software (v 3.16.6). Unadjusted prevalence and intensity of infection, with 95% confidence interval (CI), were calculated using the binomial exact test. The chi-square test was used to test the association between schistosomiasis prevalence and sex, age and freshwater contact.

For the Practical Assessment, the key unit of analysis is the prevalence per primary sampling unit (PSU = school/community), not the overall mean survey prevalence. Consequently, the results of a Practical Assessment are interpreted based on the number of PSU with a site-level prevalence greater than 10% (Fig 1) [11]. To do this, the prevalence of schistosomiasis within each PSU was calculated as follows:

$$\text{Prevalence in PSU} = \frac{\text{Number of SAC testing positive for schistosomiasis in PSU}}{\text{Total number of SAC with a valid test result in the PSU}}$$

In districts where both schistosomiasis species were present, the results were combined, so that a child positive for *S. mansoni* and/or *S. haematobium* was considered "schistosomiasis positive" and must appear once in the numerator and in the denominator. A similar calculation was performed for the combined prevalence of STH. Once the prevalence of schistosomiasis has been calculated for each PSU, the SPPA decision tree (Fig 1) was used to identify the appropriate programmatic response [11].

The feasibility of the Practical Assessment was evaluated through feedback forms, designed to assess four components of the Practical Assessment process. (One) Experience in determining whether to commence with Practical Assessments compared to Precision Assessments. This form was independently completed by the principal investigator and co-investigators directly involved in the decision to start with Practical Assessments for each district. (Two) A feedback form on the implementation of the Practical Assessment. This was completed by the field team leader and the supervisor and collected information on the ease of implementing the Practical Assessment, including site selection (schools), travel time, the number of days required to complete a site, the selection of children in schools, and sample collection. (Three) Programmatic cost of the Practical Assessment approach. This form was administered to the logistics manager to gather information on the cost and quantity of materials needed for the implementation of the SPPA approach. (Four) Interpretation of Practical and Precision Assessment results. This final form was completed by the NTD programme manager and investigators to evaluate the ease of making treatment decisions according to the SPPA decision tree.

## Results

### Schistosomiasis socio-demographic characteristics

The pilot study was conducted in May 2024 and 75 schools were visited in all five health districts studied. Four of the 75 schools initially systematically selected were replaced, including three which were closed down at the time of the study and one case of community reluctance in providing samples. Replacement was conducted as per the protocol, which called for replacement with the nearest neighbouring school. Regarding the children, 2325 out of 2400 target sample size school-age children (96.6%) were surveyed. The target sample size fell short by 75 children due to an insufficient number of target children (10–14 years) in 10 schools (two in Lola and eight in Beyla). Each of the 2325 children who participated in the survey provided stool and urine samples (Tables 1 and 2).

More boys were sampled than girls (1242/1083). The combined prevalence of both schistosomiasis species in boys and girls was 66.5% (95% CI: 63.6%-69.3%) and 66.4% (95% CI 63.7%-69.1%), respectively. The average age was 11.8 years (IQR ± 1.4 years). There was no significant difference between the prevalence of schistosomiasis according to sex or age (Table 1). Among the 2325 school-age children, 75% and 53.8% of children reported being in contact with a freshwater body during the week preceding the survey by either bathing, swimming and playing or fishing, respectively. No significant difference was observed in the prevalence of schistosomiasis between children who had been in contact with a freshwater body and those who had not.

### Schistosomiasis prevalence by health district

For both species of schistosomiasis, the prevalence of *S. mansoni* was the highest at 66.1% (95% CI: 64.1%-68.0%) (Table 2), with the majority of infections having heavy intensity compared with light intensity (46.0% vs. 20.2%). The

**Table 1. Schistosomiasis prevalence in school-aged children by sex, age and contact with water in N'Zérékoré Region, Guinea.**

| Indicators | Number of children (n = 2325) [%] | Combined SCH prevalence (95% CI) | Chi-2 |
|---|---|---|---|
| Sex | | | p-value |
| Female | 1083 [45.6%] | 66.5% (63.6-69.3) | 0.977 |
| Male | 1242 [53.4%] | 66.4% (63.7-69.1) | |
| Average age = 11.8 years (± 1.4) | | | |
| 10 years | 602 [25.9%] | 60.0% (55.9 -63.9) | 0.081* |
| 11 years old | 465 [20.0%] | 65.2% (60.6- 69.5) | |
| 12 years old | 493 [21.2%] | 71.8% (67.6 -75.7) | |
| 13 years old | 382 [16.4%] | 73.6% (68.8-77.9) | |
| 14 years old | 383 [16.5%] | 64.2% (59.2-69.0) | |
| Bathe, swim or play in nearby rivers, streams or ponds in the last week | | | |
| Yes | 1744 [75.0%] | 67.5% (65.2-69.7) | 0.725 |
| No | 581 [25.0%] | 63.3% (59.3-67.3) | |
| Fishing in nearby rivers, streams or ponds in the last week | | | |
| Yes | 1252 [53.8%] | 66.7% (64.0-69.3) | 0.066 |
| No | 1073 [46.2%] | 66.2% (63.3-69.0) | |

*Age was grouped into two categories [10–12 years] and [13–14 years] before applying the Chi-square test.

**Table 2. Prevalence of schistosomiasis among school-age children in N'Zérékoré Region, Guinea.**

| District | N | N Sm positive | Prevalence Sm (95% CI) | N Sh positive | Prevalence Sh (95% CI) | N Any SCH positive | Prevalence Any SCH (95% CI) |
|---|---|---|---|---|---|---|---|
| **Overall** | **2325** | **1537** | **66.1% (64.1-68.0)** | **101** | **4.3% (3.5-5.2)** | **1545** | **66.4% (64.5-68.4)** |
| Beyla | 435 | 244 | 56.1% (51.3-60.8) | 5 | 1.1% (0.4-2.7) | 245 | 56.3% (51.5-61.0) |
| Guéckédou | 480 | 423 | 88.1% (84.9-91.0) | 26 | 5.4% (3.6-7.8) | 425 | 88.5% (85.3-91.2) |
| Lola | 450 | 210 | 46.7% (42.0-51.4) | 4 | 0.9% (0.2-2.3) | 211 | 46.9% (42.2-51.6) |
| N'Zérékoré | 480 | 316 | 65.8% (61.4-70.1) | 5 | 1.0% (0.3-2.4) | 316 | 65.8% (61.4-70.1) |
| Yomou | 480 | 344 | 71.7% (67.4-75.7) | 61 | 12.7% (9.9-16.0) | 348 | 72.5% (68.3-76.4) |

Abbreviations: N, number of children sampled; *Sm*, S. mansoni; *Sh* S. haematobium; Any SCH, schistosomiasis, meaning combined *S. mansoni* and *S. haematobium* prevalence.

prevalence of *S. haematobium* was low at 4.3% (95% CI: 3.5%-5.2%), with the majority of these being low intensity, compared to high intensity (3.6% vs. 0.8%). The most affected health district was Guéckédou with an overall prevalence for *S. mansoni* of 88.1% (95% CI: 84.9%-91.0%), of which 66.9% had heavy intensity infections (95% CI:62.5%-71.1%). The greatest prevalence of *S. haematobium* was in Yomou with 12.7% prevalence (95% CI: 9.9%-16.0%).

The prevalence of combined schistosomiasis species was 66.4% (95% CI: 64.5%-68.4%) across all five health districts in the N'zérékoré Region. Of the 75 schools visited, 73 (97.33%) had a prevalence equal to or greater than the 10% threshold used for decision-making on PC (Fig 1). When a 50% threshold was used, 71% (52/75) of schools had a combined schistosomiasis prevalence above this threshold. Only two schools had a prevalence below the 10% threshold (in Beyla and Lola) (Fig 2A).

## Soil-transmitted helminthiasis

The combined prevalence of STH species for the five health districts was 11.7% (Table 3). By health district, combined STH prevalence ranged from 1.4% in N'zérékoré to 19.8% in Guéckédou. The predominant species was *A. lumbricoides*

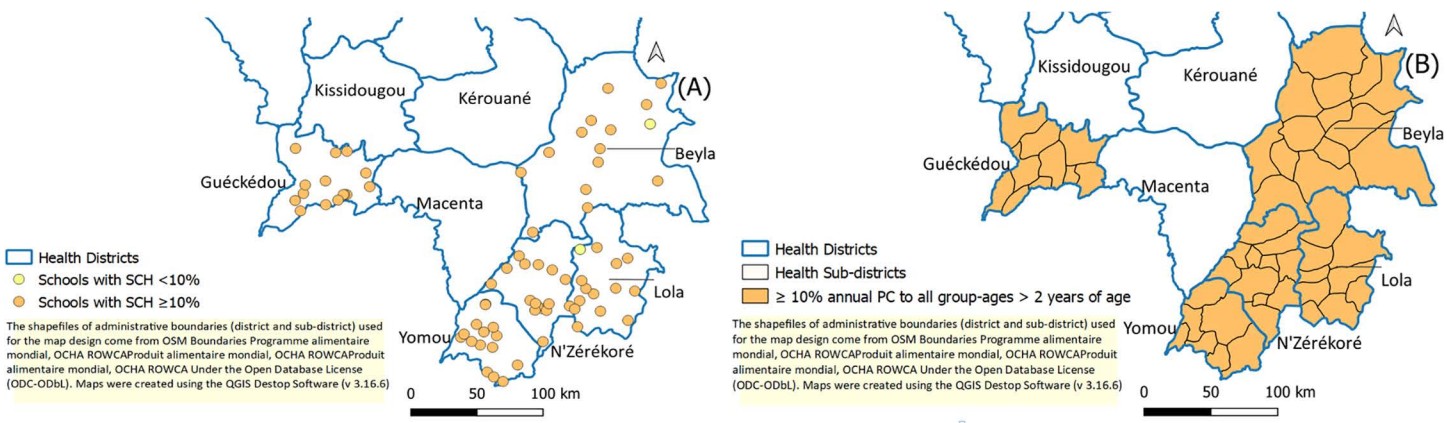

**Fig 2. SCH prevalence by school (A) and WHO PC-decision (B).**

**Table 3. Prevalence of STH in school-age children by health district in N'Zérékoré Region, Guinea.**

| District | N children examined | N positive and (prevalence) *A. lumbricoides* | N positive and (prevalence) hookworm | N positive and (prevalence) *T. trichiura* | Any STH % prevalence |
|---|---|---|---|---|---|
| **Overall** | **2325** | **215 (9.2%)** | **69 (3.0%)** | **11 (0.5%)** | **11.7%** |
| Beyla | 435 | 57 (13.1%) | 28 (6.4%) | 4 (0.9%) | 18.3% |
| Guéckédou | 480 | 100 (20.8%) | 0 (0.0%) | 1 (0.2%) | 19.8% |
| Lola | 450 | 42 (9.3%) | 17 (3.8%) | 2 (0.4%) | 12.4% |
| N'Zérékoré | 480 | 3 (0.6%) | 0 (0.0%) | 4 (0.8%) | 1.4% |
| Yomou | 480 | 13 (2.7%) | 24 (5.0%) | 0 (0.0%) | 7.1% |

Abbreviations: N, number of; Any STH, meaning combined *A. lumbricoides*, hookworm and *T. trichiura* prevalence.

with an overall prevalence of 9.2%. By health district, A. lumbricoides prevalence ranged from 0.6% in N'zérékoré to 20.8% in Guéckédou. For each species of STH, no high or moderate intensity prevalence was observed, except for *A. lumbricoides* in the Guéckédou health district, with a high intensity prevalence of 0.6%

## Feasibility of the practical assessment

The feasibility questionnaires of the Practical Assessment observed that, when following the SPPA guidance, the National PC-NTD Control Programme was easily able to use historical mapping data from 2014, coverage data, number of sub-districts and risk factors to determine whether a Practical or Precision Assessment was required in each of the health districts, and used this information to confidently select the Practical Assessment approach (S1–S4 Tables). The guidance and tools available allowed for an uncomplicated experience in the selection of the 15 schools through systematic sampling for the Practical Assessment and resulted in geographically representative sites to be sampled. Similarly, the guidance to support random selection of approximately 30 children in each school was carried out without difficulty. It took, on average across all schools, three hours per school for the process of obtaining consent from parents and then the school staff to support gathering, random selection, and readying of the children for sample collection. The training, as per the SPPA guidance, was planned for three days and was successfully completed. However, on evaluation, the teams recommended a duration of four days in total to allow local technicians to better refresh on the required laboratory techniques. Each team was composed of four people: two laboratory technicians, one supervisor, and one data enumerator. In total, 10 teams were deployed,

with two teams per health district and an average of 8.4 days were needed to complete the 15 schools in a health district. Only one team managed to conduct sampling two schools in a single day and it was deemed that reducing the number of school-age children sampled per school would feasibly allow for increasing the number of schools sampled per day to increase overall efficiency of the survey. The programmatic cost data collected as part of the feasibility evaluation are currently being analysed along with cost data from all of the SPPA pilots. Finally, in terms of interpreting the results of the survey for programmatic use, the National PC-NTD Control Programme considered the SPPA guidance and tools available to be straightforward to use and enabled clear decisions for target populations and frequency of PC in each of the sub-districts.

## Discussion

The Republic of Guinea commenced MDA campaigns using PZQ and albendazole or mebendazole nationwide in 2016. These efforts followed the severe constraints imposed by the 2014–2016 Ebola epidemic, which significantly disrupted health systems. This study provides the first data since the 2014 baseline mapping and demonstrates the prevalence and intensity of schistosomiasis and STH. The study results also offer the first precise data to support accurate and reliable sub-district level decision-making for a programmatic treatment strategy aligned with the 2022 WHO guidelines for Guinea's forest region [3].

Despite five to eight MDA campaigns since 2016, schistosomiasis and STH remain significant public health challenges. Schistosomiasis prevalence remains homogeneously high, with 97% of sampled schools reporting prevalence above 10%. These findings align with earlier studies in the region prior to the national MDA implementation [13,15,18]. The predominant species remains *S. mansoni*, consistent with data from neighbouring regions in Côte d'Ivoire, Sierra Leone, and Liberia [19–24]. Although inter-school prevalence variations were observed, the majority of surveyed schools exceeded the 10% threshold for PC eligibility, underscoring the need for continued annual treatment and expanded community-wide coverage to include all individuals aged two years and older. Because all five health districts observed >8 schools exceeding the 10% prevalence threshold, a Precision Assessment was not warranted in any of the sites. The findings indicate that the health district remains the most appropriate implementation unit; it is not necessary to transition to a sub-IU treatment strategy at this time.

Our analysis found no significant differences in prevalence or intensity by sex or age group, aligning with earlier findings from the N'Zérékoré region [15]. This contrasts with studies from Tanzania, which reported higher prevalence in males and older age groups [25]. The age range of 10–14 years in this study limited detection of broader demographic patterns. Nevertheless, the majority of studies show a prevalence and intensity of infection that peaks in 10–14 year olds, who are generally more accessible for sampling and serve as a reliable proxy for community-level infection [26–30].

In overall prevalence of schistosomiasis, we observed a decrease compared to the 2014 mapping [12] in three health districts of Beyla (66.2% vs. 56.3%), Lola (79.2% vs. 46.9%) and N'zérékoré (77.6% vs. 65.8%). In contrast, prevalence increased in the districts of Guéckédou (76.8% vs. 88.5%) and Yomou (70% vs. 72.5%) warranting further investigation. In view of the results of this study and with reference to the SPPA decision tree (Fig 1) [11], each district had between 8 and 15 schools with prevalence above the 10% threshold. In line with the first recommendation of the 2022 WHO guidelines on the control and elimination of human schistosomiasis, the National PC-NTD Control Programme must conduct annual mass treatment with PZQ in each implementation sub-unit (sub-district) [3]. When data are aggregated, four districts have prevalence above 50% and have had less than a third relative reduction in prevalence since the 2014 mapping [2,3]. This raises concerns about the lack of an adequate response in these districts and highlights the limitations of the current schistosomiasis PC strategy, which has targeted only children aged 5–14 years. Thus, at-risk individuals in other age-groups may constitute a reservoir for reinfection of freshwater bodies, promoting the parasite lifecycle and reinfection of treated children [3,31].

The WHO recommends that national programmes consider biannual (twice a year) PC instead of annual PC in endemic communities where the prevalence of *Schistosoma* infection is ≥ 10% and there is a lack of adequate response (a third relative reduction) to annual PC, despite adequate treatment coverage (≥ 75%) [2,3]. Thus, these areas of persistent infection in N'zérékoré Region require further investigation as to the drivers of high infection. Although national data reported PC coverage exceeding 90% in the N'zérékoré region, only 52.2% of surveyed children during the Practical Assessment recalled receiving PZQ during the September 2023 campaign. Independent coverage evaluation surveys are essential to verify PC reach and effectiveness, as well as risk factors associated with treatment coverage, and assessments [32]. Unfortunately, no mass treatment coverage evaluation surveys for SCH and STH have been conducted in Guinea. The reported figures for mass treatment coverage are administrative data derived from health centre and Health District Directorate (DPS) reports following mass treatment campaigns. It is strongly recommended that coverage evaluation surveys be implemented in the N'zérékoré Region before annual or biannual PC is initiated to ensure a comprehensive and tailored strategy to achieve high coverage can be planned.

An important area of ongoing research is to understand whether PZQ resistance may play a role in the suboptimal impact of PC. The Guinean team is collaborating with the Texas Biomedical Research Institute to look at markers of PZQ resistance among miracidia collected from a subset of samples used in this study. Other contributing factors include limited access to at least basic water sources, inadequate sanitation, and cultural practices that perpetuate reinfection cycles [33,34]. Addressing these issues through intersectoral collaboration and integrated control measures is essential for sustainable progress [33–35].

Interestingly, while the prevalence of schistosomiasis remains homogeneously high, the prevalence of STH has significantly declined. In Beyla, the prevalence fell from 50.2% to 18.3%, in Guéckédou from 67.6% to 19.8%, in Lola from 28.8% to 12.4%, in N'zérékoré from 54.7% to 1.4% and in Yomou from 17.2% to 7.1%. This has been observed in Mali in the Kayes region after a decade of MDA, in Liberia, and in Senegal [22,28,36]. This success reflects the positive impact of long-standing integrated efforts targeting multiple NTDs with biannual distribution of albendazole/mebendazole in primary schools and during national vaccination days, and since 2014, as part of high coverage mass treatment for lymphatic filariasis [13]. Such results highlight the potential for sustained and coordinated interventions to achieve meaningful reductions in disease burden.

However, with regards to prevalence of hookworm infection and overall combined STH prevalence, a limitation of our survey is that not all slides were read within 60 minutes of preparation, which is necessary to detect all hookworm eggs. This was due to unfavourable conditions for reading the slides in some schools and subsequent transportation to the laboratory of the nearest health centre. This limitation has likely resulted in an underestimate of overall STH prevalence, particularly in Beyla health district, where the baseline prevalence of hookworm was highest.

Based on the recommendations of the WHO monitoring and evaluation framework for schistosomiasis and soil-transmitted helminthiases [2], and following this impact assessment, the findings are as follows: i)- Beyla, Guéckédou, and Lola show a prevalence of STH infections ranging between ≥10% and <20%. Therefore, these areas will receive annual PC for five years, followed by another impact assessment. ii)- Yomou has an STH prevalence between 2% and <10% and will receive PC every two years, also followed by another impact assessment. iii)- N'Zérékoré, where the STH prevalence is below 2%, will see the suspension of PC targeting entire at-risk groups. However, drug distribution will continue in appropriate settings, such as selected child-health visits, specific school years, or antenatal care visits.

This pilot study has evaluated the feasibility of the SPPA approach and demonstrates its value as a monitoring tool. By providing actionable data at the sub-district level, the SPPA enables more precise programmatic decisions, optimising resource allocation and targeting. The integration of SPPA findings with broader public health initiatives will be critical in advancing control and elimination goals.

 

## Conclusion

The SPPA approach piloted in Guinea is a practical and effective method for evaluating MDA impact at the health district and sub-district level. The study's findings confirm homogeneously high schistosomiasis prevalence across all five health districts, with 97% of the sites exceeding the 10% threshold. The National PC-NTD Programme should sustain and strengthen annual PC while extending treatment to all age groups in alignment with WHO recommendations. Given limited reductions in schistosomiasis prevalence despite multiple MDA rounds, implementing monitoring tools such as coverage evaluation surveys is critical to ensure high treatment coverage. Advocacy with national and local authorities is essential to extend treatment coverage, raise awareness, and involve community leaders. Comprehensive control strategies combining PC, behaviour change communication, environmental management, and improved WASH services are necessary to reduce transmission and achieve disease control goals. This assessment underscores the importance of tailored programmatic responses and highlights opportunities to enhance Guinea's NTD control strategies, ultimately contributing to improved health outcomes in endemic regions.

## Supporting information

**S1 Table. Experience in determining practical assessments compared to precision assessments (Evaluation 1).** (DOCX)

**S2 Table. Feedback form on the implementation of practical evaluation (Evaluation 2).** (DOCX)

**S3 Table. Programmatic cost form for a practical assessment approach (Evaluation 3).** (DOCX)

**S4 Table. Interpretation form for practical and precision assessment results (Evaluation 4).** (DOCX)

## Acknowledgments

The authors would like to thank all the contributors who facilitated this study, in particular the National Directorate of Epidemiology and Disease Control (DNELM), the Regional Health Inspectorate Regional Health Inspectorate (RHI) and the Directorate Health District of N'Zérékoré. We very much appreciated advice on an early draft by Ashley Preston and statistical advice from Pancy Poon and Joseph Timothy (Unlimit Health). We are also grateful to the administrative, health, and educational authorities (national and regional) who provided the implementation facilities.

## Author contributions

**Conceptualization:** Balla Moussa Keita, Mamadou Diallo, Mariam Diarra, Penelope Vounatsou, Stella Kepha, Katherine Gass, Rachel L Pullan, Fiona M Fleming, Nouhou Konkoure Diallo, Mandy Kader Konde.

**Data curation:** Balla Moussa Keita, Mamadou Diallo.

**Formal analysis:** Balla Moussa Keita.

**Funding acquisition:** Balla Moussa Keita, Mandy Kader Konde.

**Investigation:** Balla Moussa Keita, Mamadou Diallo, Mariam Diarra, Souleymane Traore, Antoine Tamba Kamano, Dienabou Keita, Ablam Amento, N'falaye Kante, Alber Dopavogui, Cheick Mouctar Sylla, Nouhou Konkoure Diallo, Mandy Kader Konde.

**Methodology:** Balla Moussa Keita, Mamadou Diallo, Mariam Diarra, Penelope Vounatsou, Stella Kepha, Katherine Gass, Rachel L Pullan, Fiona M Fleming, Nouhou Konkoure Diallo, Mandy Kader Konde.

**Project administration:** Ablam Amento, Sekou Berete.

**Supervision:** Balla Moussa Keita, Mamadou Diallo, Mariam Diarra, Souleymane Traore, Antoine Tamba Kamano, Dienabou Keita, Ablam Amento, Moussa Sylla, Alber Dopavogui, Cheick Mouctar Sylla, Penelope Vounatsou, Stella Kepha, Katherine Gass, Rachel L Pullan, Fiona M Fleming, Nouhou Konkoure Diallo, Mandy Kader Konde.

**Visualization:** Balla Moussa Keita, Fiona M Fleming.

**Writing – original draft:** Balla Moussa Keita, Mamadou Diallo, Mariam Diarra, Souleymane Traore, Antoine Tamba Kamano, Nouhou Konkoure Diallo, Mandy Kader Konde.

**Writing – review & editing:** Balla Moussa Keita, Katherine Gass, Rachel L Pullan, Fiona M Fleming.

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
