## [Decision Letter · Decision Letter 0]

3 Mar 2025

Piloting the Schistosomiasis Practical and Precision Assessment approach in five health districts of the N'zérékoré region, Republic of Guinea

Dear Dr. Keita,

Thank you for submitting your manuscript to PLOS Neglected Tropical Diseases. After careful consideration, we feel that it has merit but does not fully meet PLOS Neglected Tropical Diseases's publication criteria as it currently stands. Therefore, we invite you to submit a revised version of the manuscript that addresses the points raised during the review process.

Please submit your revised manuscript within 60 days May 02 2025 11:59PM. If you will need more time than this to complete your revisions, please reply to this message or contact the journal office at plosntds@plos.org. Please include the following items when submitting your revised manuscript:

We look forward to receiving your revised manuscript.

Kind regards,

Aysegul Taylan Ozkan, M.D., Ph.D.,

Academic Editor

Jong-Yil Chai

Section Editor

Shaden Kamhawi

co-Editor-in-Chief

Paul Brindley

co-Editor-in-Chief

**Journal Requirements:**

Please ensure that the funders and grant numbers match between the Financial Disclosure field and the Funding Information tab in your submission form. Note that the funders must be provided in the same order in both places as well.

**Reviewers' Comments:**

Reviewer's Responses to Questions

**Key Review Criteria Required for Acceptance?**

**Methods** :

-Are the objectives of the study clearly articulated with a clear testable hypothesis stated?

-Is the study design appropriate to address the stated objectives?

-Is the population clearly described and appropriate for the hypothesis being tested?

-Is the sample size sufficient to ensure adequate power to address the hypothesis being tested?

-Were correct statistical analysis used to support conclusions?

-Are there concerns about ethical or regulatory requirements being met?

Reviewer #1: The method of this study was very comprehensive and explained very well. The number of sample size was adequate to increase study power.

Since the chi-square results were not significant, have the authors considered to use binomial logistic regression? Other suggestion would be to calculate (relative) risks.

Reviewer #2: The researchers aim to assess the current status of schistosomiasis infection and the treatments available for these parasites in selected health districts. They also propose evaluating the feasibility of a practical assessment. To achieve this, they conducted a descriptive cross-sectional study using a cluster sampling method, focusing on schoolchildren aged 10 to 14, and established a sample size.

However, it would be beneficial to clarify the reasons for excluding younger children from the study, as this could affect the evaluation of parasitic prevalence. While obtaining consent from the parents of younger students may be challenging, including this population could provide valuable information for a more comprehensive understanding of the studied population.

Additionally, the feasibility assessment is relevant and interesting. The researchers should present their results as they align with some of the study's objectives. If there are reasons why the results cannot be disclosed, the researchers should clarify and justify this decision.

Reviewer #3: -Are the objectives of the study clearly articulated with a clear testable hypothesis stated?

Yes! The study aims to determine the current infection status and treatment strategy for each health sub-district and to evaluate the feasibility of the impact assessment approach. While the objectives are clearly articulated, a clear testable hypothesis is not explicitly stated. It would be beneficial to include a specific hypothesis to guide the study.

-Is the study design appropriate to address the stated objectives?

The cross-sectional study design among children aged 10 to 14 years is appropriate for addressing the stated objectives. The systematic sampling of schools and random selection of children within schools ensures a representative sample. However;

o Practical assessment is the first step in the sampling process, requiring the random selection of schools across multiple subdistricts within an evaluation unit (district). The evaluation unit for this first step should be clearly defined as a health district. However, lines 139-141 seem to indicate that the subdistrict is the evaluation unit.

o The definition of implementation unit within the context of Guinea should be correctly defined. Line 155-157 seems quite confusing. Current WHO advice is for countries to transition from district to subdistrict treatments. The latter should be the new implementation unit for MDA and evaluation unit for disease specific assessments. The context in Guinea should be clearly defined.

o How many teams were formed during data collection?

o Was the evaluation unit and sampling approach for SCH identical with STH? If so, highlight reason, if no, provide justification.

-Is the population clearly described and appropriate for the hypothesis being tested?

Yes- The population is clearly described, focusing on school-aged children in five health districts of the N'zérékoré region. The inclusion and exclusion criteria are well stated.

-Is the sample size sufficient to ensure adequate power to address the hypothesis being tested?

The sample size of 2400 children is sufficient to ensure adequate power to address the hypothesis being tested. The high response rate (96.9%) further supports the robustness of the sample.

-Were correct statistical analysis used to support conclusions?

The statistical analyses used, including the chi-square test and binomial exact test, are appropriate for the data and support the conclusions drawn. The use of R software for data analysis and QGIS for mapping is suitable.

-Are there concerns about ethical or regulatory requirements being met?

Ethical considerations are thoroughly addressed, with approval from the National Committee for Health Research Ethics and informed consent obtained from village chiefs, parents, or legal guardians, and assent from children.

**Results**

-Does the analysis presented match the analysis plan?

-Are the results clearly and completely presented?

-Are the figures (Tables, Images) of sufficient quality for clarity?

Reviewer #1: The result section was greatly presented. However, there was a mistake in Table 1, in which 1,537 should be written instead of 153 in number of positive Schistosomiasis. Moreover, Figure 2 was a bit blurry, the authors might want to consider enhancing the quality of the figure.

Reviewer #2: Additionally, the feasibility assessment is relevant and interesting. The researchers should present their results as they align with some of the study's objectives. If there are reasons why the results cannot be disclosed, the researchers should clarify and justify this decision.

The rest of the results explained goodness in the first part of the objectives.

Reviewer #3: -Does the analysis presented match the analysis plan?

The analysis presented is concise though a plan wasn’t presented. However, the results are clearly and completely presented, with detailed tables and figures that enhance understanding.

-Are the results clearly and completely presented?

The results are clearly presented, with comprehensive tables showing prevalence rates and confidence intervals. The figures are of high quality and provide clear visual representations of the data. However:

o The caption in Line 230 “Schistosomiasis prevalence by health district and health sub-district” doesn’t reflect the outputs. No details or highlights of subdistrict results have been provided

o The SPPA method requires precision assessment upon completion of practical assessment. It should be clearly stated why precision assessment may not be required in this context. I presume it could be due to the homogenously high prevalence of Schisto.

o Might be worth verifying if Lola health district qualifies for precision assessment.

o A correlation between SCH infection intensity and micro/macro haematuria could be very useful in determining the extent of clinical morbidity associated with SCH.

-Are the figures (Tables, Images) of sufficient quality for clarity?

• The tables are of sufficient quality for clarity. They effectively illustrate the prevalence of SCH-STH across the health districts surveyed. However, figures were cited but not provided, so it’s hard to conduct any appraisal of quality.

**Conclusions**

-Are the conclusions supported by the data presented?

-Are the limitations of analysis clearly described?

-Do the authors discuss how these data can be helpful to advance our understanding of the topic under study?

-Is public health relevance addressed?

Reviewer #1: The conclusion was greatly supported by the data. Limitations of this study were meticulously explained in the discussion, along with the recommendations for future studies. The authors also have successfully addressed Schistosomiasis as public health issue, since the prevalence of this is still high in Republic of Guinea.

Reviewer #2: Another weakness in the choice of age group for schoolchildren was the methods used to determine the presence of parasites in feces. It should be noted that there are currently better tests for searching for parasites in feces, but they are not feasible in the reality of the study site. However, this could ideally contribute to determining more accurately the presence of parasites in feces.

An essential aspect of the discussion lacking depth is considering external factors not assessed in the study. These factors, such as different age groups, health issues, medication availability, and resistance to antiparasitics, were mentioned but not explored. It would be beneficial to compare these factors with additional literature.

The conclusions and recommendations are well-articulated and align with a public health approach. Given the significance of the results for the research site, this study presents an opportunity for future interventions based on its findings.

Reviewer #3: -Are the conclusions supported by the data presented?

The conclusions are well-supported by the data presented. The study finds that schistosomiasis remains homogenously high across all five health districts, justifying the recommendation for continued annual treatment and expanded community-wide coverage.

-Are the limitations of analysis clearly described?

The limitations of the analysis are clearly described, including the potential impact of not reading all slides within 60 minutes for detecting hookworm eggs.

-Do the authors discuss how these data can be helpful to advance our understanding of the topic under study?

The authors discuss how these data can advance our understanding of schistosomiasis control and the effectiveness of mass drug administration (MDA) programs. However:

o No conclusions or recommendations were made with regards to continuity or pause of STH MDA based on the survey findings.

o PZQ resistance was cited as a possible reason for the persistent high SCH prevalence. Before making such presumption, it is worth asking the following questions:

1. How often are coverage evaluation surveys conducted, and do these districts attain the min WHO threshold of 75% during post MDA coverage assessment?

2. What's the status of WASH infrastructure in the area?

3. What are the behavioural risk exposures to SCH infection and what are the possible SBCC initiatives tailored to the context of N’zerekore Region?

-Is public health relevance addressed?

The public health relevance is addressed, emphasizing the need for continued and expanded treatment efforts to control SCH and STH.

**Editorial and Data Presentation Modifications?**

Reviewer #1: Introduction

1. Line 66-67: please provide the correct citation and reference.

2. Line 69-71: please provide the correct citation and reference

3. Make sure to always write the full writings before using abbreviations

Results

4. Table 1: please recheck and rewrite the correct data as this is very important

5. Line 243: the authors are subjected to only put the figure once as the figure was first mentioned in the manuscript

Reviewer #2: Minor Revision

Reviewer #3: o Consider explicitly stating a clear testable hypothesis in the methods section to guide the study.

o Ensure consistency in the use of terms such as "school-age children" and "school-aged children."

o Minor grammatical corrections for improved readability, such as "school-age children" instead of "school-aged children."

o Sociodemographic information should be presented before the SCH and STH results.

**Summary and General Comments**

Reviewer #1: Piloting the Schistosomiasis Practical and Precision Assessment approach in five health districts of the N'zérékoré region, Republic of Guinea by Keita et al is really comprehensive in order to address public health issue of Schistosomiasis infection. The existence of this study really shed a light in the region to update the data from previous study conducted in 2014. The significance of this study could be directly seen in discussion session where the authors provided recommendations for all public health practitioners and stakeholders in order to improve nation's health. Please consider the inputs from the editor and reviewers to revise the manuscript so that this study could be published.

Reviewer #2: (No Response)

Reviewer #3: (No Response)

PLOS authors have the option to publish the peer review history of their article (what does this mean? ). If published, this will include your full peer review and any attached files.

**Do you want your identity to be public for this peer review?** For information about this choice, including consent withdrawal, please see our Privacy Policy .

Reviewer #1: **Yes: ** Surya Adhi, M.D.

Reviewer #2: No

Reviewer #3: No

**Figure resubmission:**

**Reproducibility:**



---

## [Decision Letter · Decision Letter 1]

30 Jul 2025

Dear Training and Research Manager Keita,

We are pleased to inform you that your manuscript 'Piloting the Schistosomiasis Practical and Precision Assessment approach in five health districts of the N'zérékoré region, Republic of Guinea' has been provisionally accepted for publication in PLOS Neglected Tropical Diseases.

Best regards,

Aysegul Taylan Ozkan, M.D., Ph.D.,

Academic Editor

Jong-Yil Chai

Section Editor

Shaden Kamhawi

co-Editor-in-Chief

Paul Brindley

co-Editor-in-Chief

Reviewer's Responses to Questions

**Key Review Criteria Required for Acceptance?**

**Methods**

-Are the objectives of the study clearly articulated with a clear testable hypothesis stated?

-Is the study design appropriate to address the stated objectives?

-Is the population clearly described and appropriate for the hypothesis being tested?

-Is the sample size sufficient to ensure adequate power to address the hypothesis being tested?

-Were correct statistical analysis used to support conclusions?

-Are there concerns about ethical or regulatory requirements being met?

Reviewer #1: (No Response)

Reviewer #2: I agree with the authors' response and edit.

Reviewer #3: Adequate responses were presented to my previous queries.

Reviewer #4: The study objectives are clear; design addressed the stated objectives. Sample size is sufficient and ethical requirements were met.

**Results**

-Does the analysis presented match the analysis plan?

-Are the results clearly and completely presented?

-Are the figures (Tables, Images) of sufficient quality for clarity?

Reviewer #1: (No Response)

Reviewer #2: I agree with the authors' response and edit.

Reviewer #3: Although the queries from my previous assessment have been addressed, it would be beneficial given that this survey provides baseline reassessment data for SCH and STH, to show how many districts are meeting the criteria for public health elimination. Specifically, please indicate the proportion of implementation units (IUs) achieving less than 1% heavy-intensity SCH infection and less than 2% moderate or heavy-intensity STH infection among SAC.

Reviewer #4: Yes as mentioned in response from previous reviewers though the plan was not included. Results well presented,

**Conclusions**

-Are the conclusions supported by the data presented?

-Are the limitations of analysis clearly described?

-Do the authors discuss how these data can be helpful to advance our understanding of the topic under study?

-Is public health relevance addressed?

Reviewer #1: (No Response)

Reviewer #2: I agree with the authors' response and edit.

Reviewer #3: The conclusions in the current version of the resubmitted article are well-supported by the data presented, and the findings contribute valuable insights to the field, addressing important public health issues related to SCH and STH. However, the study’s limitations were not acknowledged.

Reviewer #4: The conclusion well addresses the objectives. Coverage evaluation limitation is mentioned. The study is relevant and timely as most countries are in stage of implementing impact assessments. SPPA is well assessed and adaptable.

**Editorial and Data Presentation Modifications?**

Reviewer #1: (No Response)

Reviewer #2: Accept

Reviewer #3: There are no further comments beyond those previously highlighted which have now been addressed.

Reviewer #4: More details on previous MDA coverage and trends over years could make this clearer.

**Summary and General Comments**

Reviewer #1: Thank you for the authors that have comprehensively addressed all of the reviewer’s and editor’s comments.

Reviewer #2: Accept

Reviewer #3: (No Response)

Reviewer #4: Line 112-5-8 round of treatment. It is important to document the coverage over years and methods of ensuring good coverage is attained. Further details to this may help justifying the impact

Need to correct use of programmatic and epidemiological coverage. Line 113 should be epidemiological coverage.

PLOS authors have the option to publish the peer review history of their article (what does this mean? ). If published, this will include your full peer review and any attached files.

**Do you want your identity to be public for this peer review?** For information about this choice, including consent withdrawal, please see our Privacy Policy .

Reviewer #1: **Yes: ** Surya Adhi, MD

Reviewer #2: **Yes: ** Jorge Alave

Reviewer #3: No

Reviewer #4: No

---

## [Editor Report · Acceptance letter]

Dear Training and Research Manager Keita,

We are delighted to inform you that your manuscript, "Piloting the Schistosomiasis Practical and Precision Assessment approach in five health districts of the N'zérékoré region, Republic of Guinea," has been formally accepted for publication in PLOS Neglected Tropical Diseases.

Best regards,

Shaden Kamhawi

co-Editor-in-Chief

Paul Brindley

co-Editor-in-Chief
